# Dynamic Self-Distillation via Previous Mini-batches for Fine-tuning Small Language Models

## Abstract

Knowledge distillation (KD) has become a widely adopted approach for compressing large language models (LLMs) to reduce computational costs and memory footprints. However, the availability of complex teacher models is a prerequisite for running most KD pipelines. Thus, the traditional KD procedure can be unachievable or budget-unfriendly, particularly when relying on commercial LLMs like GPT4. In this regard, Self-distillation (SelfD) emerges as an advisable alternative, enabling student models to learn without teachers' guidance. Nonetheless, existing SelfD approaches for LMs often involve architectural modifications, assuming the models are open-source, which may not always be practical. In this work, we introduce a model-agnostic and task-agnostic method named **dyn**amic **S**elf**D** from the **p**revious mini-**b**atch (**DynSDPB**), which realizes current iterations' distillation from the last ones' generated logits. Additionally, to address prediction inaccuracies during the early iterations, we dynamically adjust the distillation influence and temperature values to enhance the adaptability of fine-tuning. Furthermore, DynSDPB is a novel fine-tuning policy that facilitates the **seamless integration** of existing self-correction and self-training techniques for small language models (SLMs) because they all require updating SLMs' parameters. We demonstrate the superior performance of DynSDPB on both encoder-only LMs (e.g., BERT model families) and decoder-only LMs (e.g., LLaMA model families), validating its effectiveness across natural language understanding (NLU) and natural language generation (NLG) benchmarks.

## 1 Introduction

Both pre-trained language models (PLMs) (Sun et al., 2022) and large language models (LLMs) (Zhao et al., 2023) [1] have shown remarkable performance across various natural language understanding (NLU) (Khurana et al., 2023) and natural language generation (NLG) (Dong et al., 2022) tasks. However, their impressive functionality is usually accompanied with the heavy computational burden brought by LLMs' abundant parameters. This can be alleviated by model compression (Wang et al., 2024b), where Knowledge distillation (KD) (Hinton et al., 2015) acts as a practical solution. Yet, existing KD techniques in both encoder-only LMs (Wang et al., 2023a; Sengupta et al., 2023) and decoder-only LMs (Zhu et al., 2023; Hsieh et al., 2023; Liu et al., 2023) all fall within the classical KD's framework that involves first pretraining large teacher models and then transferring their knowledge to small student models shown in Figure 1(a). The research on how to enhance the fine-tuning performance of small language models (SLMs) without LLMs remains relatively **unexplored**. Although nowadays LLMs can be easily accessed via API calls, yet relying on them to realize a successful KD critically requires obtaining sufficient synthetic data to help fine-tune SLMs by querying online LLMs (e.g., GPT-4 (Achiam et al., 2023)) which might be prohibitively expensive (Wang et al., 2024a). For instance, the total cost of API usage for preliminary

---

[1] In this work, **PLMs** refer to **encoder-only** LMs like BERT (Devlin et al., 2018), RoBERTa (Liu et al., 2019b), and DeBERTa (He et al., 2020; 2021). **LLMs** refer to **decoder-only** LMs with billions of parameters (e.g., Llama-3.1-70B/405B Dubey et al. (2024) while **small LMs (SLMs)** refer to **decoder-only** LMs with a few billion parameters (e.g., LLaMA-2-7/13B (Touvron et al., 2023)), followed by (Zhang et al., 2024b). **LMs** is a general term used to refer to PLMs, LLMs, and SLMs.

experiments in Fine-tune-CoT (Ho et al., 2022) amounted to 1,981 dollars. Additionally, users may confront inconvenience of queuing delays when using cloud LLMs. And, even worse, teacher LLMs may unintentionally impart their biases and unfairness to student SLMs (Gallegos et al., 2024).

To address those challenges, self-distillation (SelfD) (Zhang et al., 2019) is proposed to enable small-scale models to distill knowledge within themselves to improve testing performance. Nonetheless, conventional SelfD techniques (Zhang et al., 2019; Liu et al., 2020) require heavy architecture modifications, which is infeasible for proprietary LLMs like GPT-4 (Achiam et al., 2023). Therefore, to propose a novel SelfD method that enables effectively fine-tuning of LMs without accessing their architectures, taking inspiration **inspiration** from DLB (Shen et al., 2022), we design a customized data loading strategy and allow student PLMs or SLMs to leverage knowledge from the last mini-batch information with the purpose of boosting their fine-tuning performance. However, DLB is **static** that fails to consider the reality that students' early-stage generalization capability is weak and gradually evolving during the fine-tuning process. Moreover, unlike image classification where the models' output size is fixed, autoregressive LLMs might generate a **varying** number of output tokens even for the same input (Wang et al., 2022c), thus leading to inconsistencies in output sequence length for the same input but at different fine-tuning iterations.

Motivated by those findings, we introduce **dyn**amic **S**elf**D** from the **p**revious mini-**b**atch (**DynSDPB**), shown in Figure 1(b), with the aim at effectively fine-tuning PLMs or SLMs without LLMs. Specifically, DynSDPB realizes a novel SelfD technique via the soft targets from the latest mini-batch to guide the training of the current mini-batch, which can be applied to both encoder-only and decoder-only LMs. Moreover, DynSDPB enables students to dynamically adjust their SelfD settings (distillation factor $\alpha$ and temperature $\tau$) according to their evolving proficiency measured by prediction uncertainty and discrimination capability. Furthermore, we propose a novel method called Vocabulary Map Matching (VMM) in order to address output dimension mismatch caused by the varying number of generated tokens from autoregressive LLMs for the same input across different iterations. Lastly, DynSDPB, as a regularization form, is able to mitigate gradient vanishing when fine-tuning PLMs such as DeBERTa (He et al., 2020) shown in Figure 2. Overall, the **major contributions** of this paper are four-fold:

- To the best of our knowledge, we are the **first** to propose a SelfD method called DynSDPB to effectively fine-tune both encoder-only and decoder-only LMs via the last mini-batch's information without complex teacher models.

- DynSDPB is **adaptive** that students can dynamically adjust their fine-tuning strategy based on current states. Moreover, we introduce a novel method called Vocabulary Map Matching (VMM) to address output dimension mismatch for auto-regressive LMs.

- Our method is a **plug-in** technique that can be seamlessly integrated into to existing Self-Training/Correction methods for SLMs (Wang et al., 2024a; Zhang et al., 2024b).

- Experiments on both encoder-only PLMs (e.g., RoBERTa-base) for NLU and decoder-only SLMs (e.g., LLaMA2-7B) for NLG demonstrate the effectiveness of DynSDPB.

## 2 RELATED WORK

**KD for encoder-only LMs.** Knowledge distillation (KD) (Hinton et al., 2015) aims to transfer dark knowledge (soft labels) from large-scale teachers to smaller-scale students. Since its introduction, a large amount of work has been investigated in the area of PLMs (Sun et al., 2019; Sanh et al., 2019). Specifically, KD works about PLMs can be roughly classified into one-stage methods and two-stage ones. One-stage methods perform distillation only at the fine-tuning stage, which is task-specific (Sun et al., 2019; Sengupta et al., 2023). Two-stage methods perform distillation at both the pre-training and the fine-tuning stages, which is task-agnostic (Sanh et al., 2019; Liang et al., 2023). The detailed literature review is in Appendix A.1. In this paper, we explore a scenario where students distill knowledge within themselves instead of approximating output logits from complex teachers because sometimes they might be unavailable due to limited budgets.

**KD for decoder-only LMs.** Recently, studying KD in auto-regressive LLMs (Agarwal et al., 2024; Ko et al., 2024) have attracted researchers' attention. Furthermore, several studies have focused on leveraging the chain of thought (CoT) (Wei et al., 2022) reasoning generated by LLMs

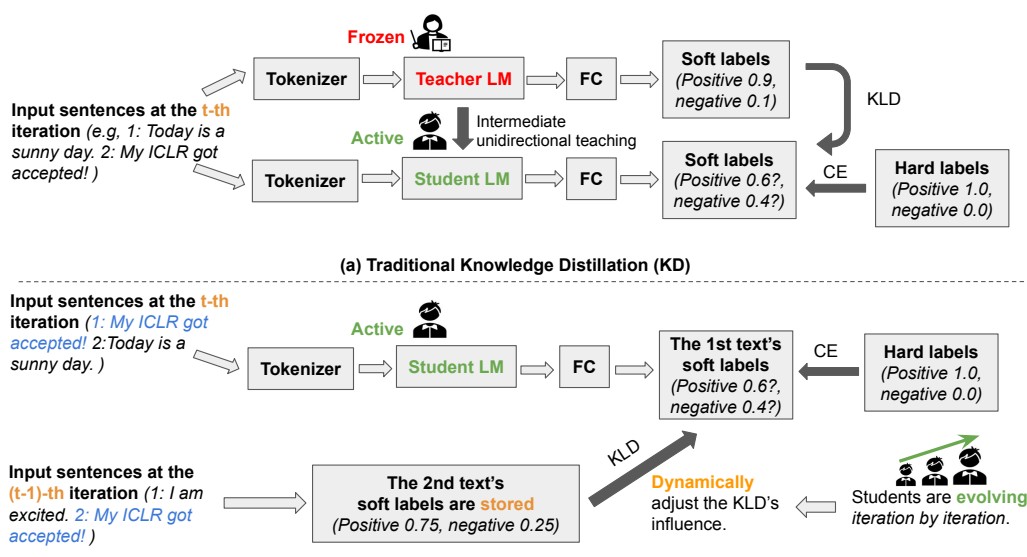

Figure 1: Two types of distillation. (a) displays the classical knowledge distillation (KD) framework that requires a teacher model. (b) outlines our **dyn**amic **S**elf**D** from the **p**revious mini-**b**atch (DynSDPB), where we just let student models distill knowledge from itself via the last iteration's information. Considering that students are evolving during distillation, we design a mechanism to **dynamically** adjust $\tau$ in Eq. (6) and $\alpha$ in Eq. (7). CE means cross-entropy, KLD means Kullback-Leibler Divergence, and FC means fully connected layers.

to enhance SLMs' reasoning abilities (Ho et al., 2022; Magister et al., 2022; Shridhar et al., 2022; Wang et al., 2023b;c; Chen et al., 2023; Fu et al., 2023; Zhu et al., 2023; Li et al., 2023; Liu et al., 2023). For instance, (Hsieh et al., 2023) introduced "Distilling step-by-step" for extracting rationales from LLMs as additional supervision for fine-tuning SLMs. However, all of these methods utilize GPT-3.5-turbo as the teacher, whose provider charges based on the total number of tokens (input + output) processed in a single API call. In contrast, our method can be executed offline on a local NVIDIA-4090 desktop without extra costs, and **seamlessly integrated** into the above KD methods since they all demand fine-tuning SLMs (e.g., LLaMA-2-7B (Touvron et al., 2023)).

**Self Distillation.** Self-distillation (SelfD) is a technique that student models learn by themselves without teachers (Furlanello et al., 2018). Most SelfD research has focused on computer vision (CV) (Zhang et al., 2019; Yun et al., 2020; Zheng & Peng, 2022), with fewer studies in natural language processing (NLP). A representative SelfD approach is Be Your Own Teacher (BYOT) (Zhang et al., 2019), which adds extra classifiers in the intermediate layers of a ResNet (He et al., 2016) to distill knowledge from deeper layers into shallower ones. Interestingly, Early Exit (EE) (Xu & McAuley, 2023) is related to BYOT, as it uses inserted classifiers in BERT (Devlin et al., 2018) for adaptive inference, with more details in Appendix A.2. However, both EE (Xin et al., 2020) and BYOT (Zhang et al., 2019) require models to be open-source to modify their architectures. To address this, DLB (Shen et al., 2022) was introduced that distills knowledge from the previous mini-batch and only changes the data loading procedure without accessing to models' structure. Despite this, DLB still has some limitations. First, since students are learning on their own, their predictions in the early stages are too inaccurate to effectively contribute to SelfD, as seen in Table 1. Second, keeping hyperparameters fixed as models evolve limits their potential to improve distillation performance (Li et al., 2021). Lastly, unlike image classification where outputs have fixed sizes (Shen et al., 2022), autoregressive LLMs can generate varying token lengths for the same input (Wang et al., 2022c), causing output length mismatch for the same text sequence but at different fine-tuning iterations.

**Self Training.** To the best of our knowledge, the **most related** work is enhancing LLMs through self-training methods (Zelikman et al., 2022; Gulcehre et al., 2023; Singh et al., 2023), where they

encourage LLMs to learn from their own generated data in a semi-supervised framework while our method is a supervised setting. Moreover, we note that there exist **orthogonal techniques** like Self-Training with DPO (Wang et al., 2024a) or Self-Correction (SCORE) (Zhang et al., 2024b) to improve SLMs without LLMs. These techniques can be seamlessly integrated into our work because they both demand fine-tuning SLMs, where we leave the integration of them to future work.

# 3 THE PROPOSED METHOD

## 3.1 PROBLEM FORMULATION

In this work, we focus on both NLU and NLG benchmarks. We denote the training dataset with $N$ instances as $\mathcal{D}_{train} = \{x_i\}_{i=1}^N$ where $x_i$ is the input sentence, and the corresponding ground truth is $\mathcal{Y}_{train} = \{y_i\}_{i=1}^N$ with $y_i \in C$ for **encoder-only PLMs** like BERT (Devlin et al., 2018) where $|C|$ is the class size, and $\mathcal{Y}_{train} = \{(y_1^i, y_2^i, \ldots, y_m^i)\}_{i=1}^N$ with token $y_j^i \in V$ for **causal decoder-only LLMs** like LLaMA-2 (Touvron et al., 2023) where $|V|$ is the vocabulary size. For **NLU** tasks, the raw sentence $x_i$ is transformed into a contextualized representation $h_i^{cr} = \text{EncoderLM}(x_i)$. A softmax layer with a learnable parameter tensor $W$ is then appended for producing **soft labels** $p_i = \text{softmax}(h_i)$ defined in Eq. (2), where $h_i = W \cdot h_i^{cr}$ are termed as **output logits**. At each training iteration, a mini-batch of $n$ samples $\mathcal{B} = \{(x_i, y_i)\}_{i=1}^n \subseteq \mathcal{D}_{train}$ are randomly sampled and are fed into a target LM parameterized by $\theta$ to optimize the cross-entropy (CE) loss function:

$$\mathcal{L}_{\text{CE}}^\theta = -\frac{1}{n}\sum_{i=1}^n y_i \cdot log(p_i), \tag{1}$$

where $p_i = (p_i^1, \ldots, p_i^C)$ is the predictive probability distribution and for class $c \in C$:

$$p_i^c = \frac{\exp(h_i^c(x_i;\theta)/\tau)}{\sum_{j=1}^C \exp(h_i^j(x_i;\theta)/\tau)}, \tag{2}$$

where $h_i^c$ stands for the $c$-th component of the output logits, and temperature hyperparameter $\tau$ is usually 1. For **NLG** tasks, a token-level auto-regressive policy $p(.|y_{<n}^i, x_i) \in (0,1)^{|V|}$ outputs a next-token probability distribution over all tokens in $V$, conditioned on the input $x_i$ and output sequence $y_{<n}^i$. Thus, auto-regressive generation involves predicting tokens sequentially based on the previously generated tokens. The probability of predicting $n$-th token $y_n^i$, $p(y_n^i|y_{<n}^i, x_i)$, is:

$$p(y_n^i|y_{<n}^i, x_i) = \frac{\exp(z_n/\tau)}{\sum_{v=1}^{|V|} \exp(z_v/\tau)}, \tag{3}$$

where $z_n$ is the logit score for the token $y_n$. Higher values of $\tau$ introduce more randomness while a lower value makes the output more likely to be the most probable words. To improve students' generalization abilities, vanilla KD (Hinton et al., 2015) transfers pre-trained teachers' knowledge by reducing an additional Kullback-Leibler (KL) divergence loss between the soft labels from the teacher and the student in every mini-batch:

$$\mathcal{L}_{\text{KD}} = -\frac{1}{n}\sum_{i=1}^n \tau^2 D_{\text{KL}}(p_i^T \| p_i^S), \tag{4}$$

where $p_i^T$ and $p_i^S$ are soft labels smoothed by $\tau$ from the teacher and the student, respectively. Hence, the overall loss function $L_{\text{total}}^\theta$ of KD is as follows with hyperparameter $\alpha$ to balance two terms:

$$\mathcal{L}_{\text{total}}^\theta = L_{\text{CE}}^\theta + \alpha \cdot L_{\text{KD}}. \tag{5}$$

## 3.2 OUR MOTIVATIONS

**Motivation 1** Previous KD methods for LMs (Sengupta et al., 2023; Hsieh et al., 2023; Liu et al., 2023) usually rely on a complex teacher to generate $p_i^T$, which might be unavailable or infeasible to obtain due to limited computational budgets. Moreover, existing SelfD works (Liu et al., 2020; Xin et al., 2020) in Appendix A.2 assumes the given LMs are open-source so that we could modify their architecture by inserting classifiers to benefit SelfD training, which is sometimes unrealistic. To address these limitations, we utilize historical output logits from the previous mini-batch to generate proxy $p_i^T$ serving as immediate smoothed labels for self-distilling PLMs or SLMs **inspired** by DLB (Shen et al., 2022) that focuses on enhancing models' generalization on image classification.

**Motivation 2**   If we simply employ DLB in fine-tuning LMs, one concern exists that training curves are likely to be misdirected from models' incorrect predictions in the early stages (Li et al., 2021). We attribute this adversity to the fact that the original DLB framework (Shen et al., 2022) is **static**, i.e., the hyperparameters ($\tau$ and $\alpha$) are strictly fixed during the course of SelfD. In this regard, it's natural to conduct adaptive adjusting of the hyperparameter settings as student models are constantly evolving during self-teaching. Motivated by this, we explore a dynamic SelfD framework, whose core idea is to empower students to **dynamically** adjust the hyperparameters ($\tau$ and $\alpha$) based on their current iteration's generation abilities. Furthermore, both image classification and NLU tasks have fixed output dimensions, enabling students to seamlessly teach themselves for the same input across different fine-tuning iterations via Eq. (4). However, decoder-only LMs for NLG may produce a **varying** number of output tokens for the same input at different iterations (Wang et al., 2022c), resulting in **length mismatch** between output sequences, which has to be addressed.

### 3.3   METHODOLOGY

**Basic Strategy**   Our SelfD framework is visualized in Figure 1(b). Instead of adopting a complex teacher to provide guidance $p_i^T$, we utilize the backup logits from the last mini-batch to generate teaching soft targets. Formally, given that at the $t$-th iteration the model is parameterized by $\theta_t$, we substitute the $p_i^T$ and $p_i^S$ in Eq. (4) by the soft labels $p_i^{S,t-1}$ and $p_i^{S,t}$ generated by the same model parameterized by $\theta_{t-1}$ and $\theta_t$, respectively. Thus, we introduce a last-mini-batch consistency (LMBC) regularization loss defined as follows:

$$\mathcal{L}_{\text{LMBC}}^{\theta_t} = -\frac{1}{n} \sum_{i=1}^{n} \tau^2 \cdot D_{\text{KL}}(p_i^{S,t-1} \| p_i^{S,t}). \tag{6}$$

Specifically, we denote the mini-batch of data sampled at the $t$-th iteration as $\mathcal{B}_t = \{(x_i^t, y_i^t)\}_{i=1}^{n}$ and design a **special data sampler** to iteratively obtain $\mathcal{B}_{t-1}$ and $\mathcal{B}_t$, where the samples in the **right half** of $\mathcal{B}_{t-1}$ are constrained to coinciding with the ones in the **left half** of $\mathcal{B}_t$ shown in Figure 1(b). At the $t$-th iteration, we only need to save logits from $\mathcal{B}_t$ and apply them into the next iteration's regularization, requiring very few extra memory footprints. Intuitively, the target network serves **both** as a teacher and a student within each mini-batch. As a teacher, it generates soft targets to regulate itself in subsequent iterations. As a student, it absorbs smoothed labels from the previous iteration besides minimizing the CE loss. Therefore, the overall loss function is denoted as:

$$\mathcal{L}_{\text{total}}^{\theta_t} = L_{\text{CE}}^{\theta_t} + \alpha \cdot L_{\text{LMBC}}^{\theta_t}. \tag{7}$$

**Dynamic Improvement**   Note that the above SelfD framework is **static** where hyperparameters ($\tau$ and $\alpha$) are rigidly fixed all the time. Since student models are continually evolving during fine-tuning, adaptively adjusting the hyperparameter settings according to students' different states has the potential of bringing benefits into their final performance on unknown testing data. In this work, we adopt **prediction uncertainty** (Li et al., 2021) as a measurement of students' generalization capability for input data. Note that the dynamic framework in (Li et al., 2021) requires complex teachers while in this study we assume those teachers are unavailable. Formally, given $n$ instances in one mini-batch, for each instance $x_i$, we could obtain its output class probability distribution $p_i$ over $C$ classes. Once getting it, we compute an **uncertainty score** $u_{x_i}$ for $x_i$ using the following entropy formula that measures the uncertainty of the student prediction distribution:

$$u_{x_i} = -\sum_{c=1}^{|C|} p_i^c \log p_i^c \text{ for encoder-LMs, and } u_{x_i} = -\sum_{v=1}^{|V|} p_i^v \log p_i^v \text{ for decoder-LMs}, \tag{8}$$

where in the early stages $u_{x_i}$ are bigger since students are less competent. However, not only students' predictions are uncertain in the beginning, but also they generate incorrect predictions with high likelihood. Making students self-distill these incorrect soft labels is a disaster that must be avoided. To fix the model misleading issue caused by incorrect predictions, we evaluate student's **discrimination capability** $d_{x_i} = (1 + exp(-y_i \cdot log(p_i)))^{-1}$ to dynamically adjust temperature $\tau$ via being multiplied by $d_{x_i}$, where temperatures will be raised to further smooth soft targets when prediction losses are larger, and lowered to preserve more discriminative information when prediction losses are smaller. In short, we use prediction uncertainty $u_{x_i}$ and discrimination capability $d_{x_i}$

to customize the distillation importance factor $\alpha$ and temperature $\tau$ for each sample $x_i$, respectively. The modified overall loss function is thus defined as follows:

$$\tilde{\mathcal{L}}_{\text{total}}^{\theta_t} = L_{\text{CE}}^{\theta_t} + (1 - \frac{u_{x_i}}{U}) \cdot \alpha \cdot \tilde{L}_{\text{LMBC}}^{\theta_t}, \tag{9}$$

where $U$ is a normalization factor re-scaling the weight to $[0, 1]$, and the modified LMBC loss is:

$$\tilde{\mathcal{L}}_{\text{LMBC}}^{\theta_t} = -\frac{1}{n} \sum_{i=1}^{n} (d_{x_i}\tau)^2 \cdot D_{\text{KL}}(p_i^{S,t-1} \| p_i^{S,t}). \tag{10}$$

**Output Mismatch Alignment for NLG**  It's common that decoder-only SLMs can generate a varying number of output tokens for the same input at different iterations (Wang et al., 2022c). Specifically, for the same input $x_i$, the student may produce $m_{t-1}$ tokens $(y_1^i, y_2^i, \ldots, y_{m_{t-1}}^i)$ at the $(t-1)$-th iteration, and $m_t$ tokens $(y_1^i, y_2^i, \ldots, y_{m_t}^i)$ at the $t$-th iteration, where $m_{t-1} \neq m_t$, making it impossible to directly apply Eq. (4). To **align** this mismatch, we sum the token vectors within each output sequence to form a **vocabulary map**, $y_{t-1}$ or $y_t$, of dimension $|V|$, since each token $y_i$ represents a probability distribution over $|V|$ tokens. We then normalize the vocabulary maps $y_{t-1}$ or $y_t$ to the $[0, 1]$ range. Now we can substitute $p_i^{S,t-1}$ with $y_{t-1}$ and $p_i^{S,t}$ with $y_t$ in Eq. (6). We call this method Vocabulary Map Matching (**VMM**). Our **intuition** is that although the output sequence length may vary for the same input, the corresponding vocabulary maps should capture very similar semantics represented by some important tokens having higher probability.

**Algorithmic Details**  Overall, the target LM are a teacher and a student interchangeably at each fine-tuning iteration. In the teacher role, it offers soft targets to guide its next iteration. As for the student role, it distills smoothed labels produced in the previous iteration besides concentrating on minimizing the CE loss. The framework is shown in Algorithm 1 in Appendix B.

## 4 EXPERIMENTS

### 4.1 DATASETS AND SETTINGS

**Datasets**  Following the latest studies (Sengupta et al., 2023; Shi et al., 2024), we evaluate the effectiveness of DynSDPB on a wide range of tasks, including natural language understanding (NLU) and natural language understanding (NLG). A summary of datasets is presented in Appendix C.

**Implementation Details**  We use representative encoder-only PLMs (BERT (Devlin et al., 2018), RoBERTa (Liu et al., 2019b), ALBERT (Lan et al., 2019), and DeBERTa-v1/v2/v3-large (He et al., 2020; 2021)), and decoder-only SLMs (LLaMA-1-7B (Touvron et al., 2023), LLaMA-2-7B/13B (Touvron et al., 2023), and LLaMA-3-8B (Dubey et al., 2024)) as students to evaluate DynSDPB. We conduct a grid hyper-parameter search for the baseline methods and our method (DynSDPB) similar to (Sun et al., 2019). Appendix D gives more details to reproduce our experimental results.

**Baselines**  We compare our method (DynSDPB) with Finetune, Double Finetune (double training epochs compared with Finetune), and Sequential/Random DLB (whether to shuffle datasets while applying DLB (Shen et al., 2022)). Moreover, as our method is strongly related to KD, we provide a focused comparison to representative KD methods in PLMs, including vanilla KD (Hinton et al., 2015), and other competitive KD methods such as patient knowledge distillation (PKD) (Sun et al., 2019), Gradient Knowledge Distillation (GKD) (Wang et al., 2022b), KD via Knowledge Selection (Wang et al., 2023a), KD with meta learning (Meta Distill) (Zhou et al., 2021), KD by learning good teachers (LGTM) (Ren et al., 2023), and Retrieval-augmented KD (ReAugKD) (Zhang et al., 2023). It is worth noting that, although traditional KD techniques assume the presence of a complex teacher model, our method relies solely on student self-teaching, yet we still achieve results that are comparable to, or even better than, some of these approaches.

### 4.2 MAIN RESULTS

**Results on NLU Tasks**  Table 1 presents the performance results from the **GLUE** (Wang et al., 2018) benchmark obtained by RoBERTa (Liu et al., 2019b), BERT (Devlin et al., 2018), and ALBERT (Lan et al., 2019). We find that (i) All three PLMs improve performance via SelfD from

Table 1: Results on NLU tasks from the GLUE (Wang et al., 2018) benchmark. The best and second-best results are in **bold** and *italics*, respectively. RoBERTa-base$_6$ means the 6-layer version initialized by the first six layers of RoBERTa-base$_{12}$.

| Methods | Counterpart | #Params | RTE | COLA | MNLI-m/mm | SST-2 | QNLI | QQP | MRPC |
|---|---|---|---|---|---|---|---|---|---|
| Finetune | RoBERTa-base$_6$ | 83.0M | 60.6 | 52.1 | 84.2/84.0 | 92.0 | 90.5 | 91.1 | 86.5 |
| Double Finetune | RoBERTa-base$_6$ | 83.0M | 61.7 | 53.7 | 84.9/84.8 | 92.2 | 90.8 | 91.3 | 86.9 |
| Sequential DLB | RoBERTa-base$_6$ | 83.0M | 66.8 | 52.1 | 85.0/84.8 | 92.9 | 90.0 | 92.0 | 87.1 |
| Random DLB | RoBERTa-base$_6$ | 83.0M | *67.6* | *55.0* | *85.4/85.1* | *93.1* | *90.6* | *92.2* | *87.5* |
| **DynSDPB (Ours)** | RoBERTa-base$_6$ | 83.0M | **68.3** | **56.0** | **85.9/85.3** | **93.9** | **91.8** | **92.7** | **88.0** |
| Finetune | BERT-base$_6$ | 67.0M | 64.9 | 38.3 | 81.1/79.8 | 89.5 | 86.5 | 88.2 | 79.2 |
| Double Finetune | BERT-base$_6$ | 67.0M | 63.9 | 38.6 | 81.2/80.7 | 89.9 | 86.9 | 89.5 | 81.8 |
| Sequential DLB | BERT-base$_6$ | 67.0M | 66.5 | 40.4 | 81.7/81.5 | 90.5 | 87.1 | 89.9 | 81.5 |
| Random DLB | BERT-base$_6$ | 67.0M | *67.5* | *42.8* | *82.2/81.9* | *90.9* | *87.8* | *90.2* | *82.1* |
| **DynSDPB (Ours)** | BERT-base$_6$ | 67.0M | **68.2** | **43.5** | **82.7/82.2** | **91.5** | **88.4** | **91.0** | **82.6** |
| Finetune | ALBERT-base$_6$ | 6.0M | 57.8 | 48.9 | 80.9/80.8 | 91.1 | 87.1 | 87.5 | 85.3 |
| Double Finetune | ALBERT-base$_6$ | 6.0M | 61.1 | 49.4 | 82.4/82.3 | 91.4 | 87.5 | 88.9 | 85.8 |
| Sequential DLB | ALBERT-base$_6$ | 6.0M | 63.2 | 50.2 | 82.1/81.5 | 91.3 | 88.1 | 89.2 | 86.1 |
| Random DLB | ALBERT-base$_6$ | 6.0M | *65.1* | *51.1* | *83.1/82.7* | *91.9* | *88.5* | *89.7* | *86.8* |
| **DynSDPB (Ours)** | ALBERT-base$_6$ | 6.0M | **67.2** | **51.9** | **83.5/83.1** | **92.5** | **89.3** | **90.6** | **87.5** |
| Finetune | RoBERTa-base$_{12}$ | 125.0M | 64.9 | 59.6 | 87.6/87.3 | 93.1 | 91.5 | 91.4 | 88.9 |
| Double Finetune | RoBERTa-base$_{12}$ | 125.0M | 73.3 | 61.5 | 87.5/87.3 | 93.5 | 92.0 | 91.7 | 89.2 |
| Sequential DLB | RoBERTa-base$_{12}$ | 125.0M | 78.4 | 58.2 | 87.9/87.5 | 94.2 | 92.5 | 91.9 | 90.4 |
| Random DLB | RoBERTa-base$_{12}$ | 125.0M | *79.1* | *62.2* | *88.3/88.1* | *93.9* | *93.1* | *92.9* | *90.9* |
| **DynSDPB (Ours)** | RoBERTa-base$_{12}$ | 125.0M | **79.8** | **62.8** | **88.9/88.4** | **94.8** | **93.7** | **93.5** | **91.5** |
| Finetune | BERT-base$_{12}$ | 110.0M | 69.3 | 56.9 | 84.1/83.1 | 92.7 | 90.3 | 90.5 | 85.5 |
| Double Finetune | BERT-base$_{12}$ | 110.0M | 69.7 | 57.6 | 84.3/84.2 | 93.0 | 91.1 | 91.3 | 86.3 |
| Sequential DLB | BERT-base$_{12}$ | 110.0M | 70.8 | 56.3 | 84.7/84.4 | 93.3 | 91.4 | 91.9 | 87.1 |
| Random DLB | BERT-base$_{12}$ | 110.0M | *71.1* | *58.9* | *85.2/84.7* | *93.2* | *92.0* | *92.2* | *87.3* |
| **DynSDPB (Ours)** | BERT-base$_{12}$ | 110.0M | **71.9** | **59.7** | **85.9/85.1** | **94.1** | **92.8** | **92.9** | **88.5** |
| Finetune | ALBERT-base$_{12}$ | 11.0M | 68.9 | 56.1 | 76.3/76.5 | 90.5 | 89.7 | 89.5 | 88.7 |
| Double Finetune | ALBERT-base$_{12}$ | 11.0M | 70.7 | 56.5 | 84.9/84.4 | 91.1 | 90.5 | 90.4 | 88.2 |
| Sequential DLB | ALBERT-base$_{12}$ | 11.0M | 73.4 | 57.1 | 84.4/84.1 | 91.8 | 91.9 | 90.9 | 89.5 |
| Random DLB | ALBERT-base$_{12}$ | 11.0M | *74.4* | *58.2* | *85.2/84.9* | *92.1* | *92.4* | *91.5* | *90.4* |
| **DynSDPB (Ours)** | ALBERT-base$_{12}$ | 11.0M | **75.8** | **59.4** | **85.9/85.6** | **93.5** | **92.7** | **92.1** | **90.9** |

the last mini-batch compared to vanilla fine-tuning, indicated by the all positive values in the "Sequential/Random DLB" rows. (ii) The COLA (Warstadt et al., 2019), RTE (Bentivogli et al., 2009), MRPC (Dolan & Brockett, 2005) datasets having smaller sizes generally benefit more from SelfD. (iii) Applying SelfD strategy to models is still better than Double Finetune, where two methods consume the same amount of computational energy. (iv) The downstream performance is further improved if utilizing the dynamical framework to adaptively adjust $\alpha$ and $\tau$ compared to the static versions on all datasets illustrated from the "DynSDPB (Ours)" row, which can prove our method's effectiveness. Table 3 presents the validation results from the **SuperGLUE** (Wang et al., 2019) benchmark obtained by three BERT-style LLMs: DeBERTa-v1/v2-large (He et al., 2020) and DeBERTa-v3-large (He et al., 2021). Similar to results from Table 1, we find that Dynamic SelfD does also significantly improve models' generation abilities in comparison with four baseline methods on the SuperGLUE benchmark.

**Results on NLG Tasks** Table 4 presents the results on various NLG tasks. The findings align closely with those observed in NLU tasks (GLUE in Table 1 and SuperGLUE in Table 3). Compared to baseline approaches, our method demonstrates notable improvements across almost all reasoning-based tasks, thus confirming the broad effectiveness of our proposed DynSDPB approach for NLG tasks. We observe that our method achieve more remarkable results on HS than other NLG tasks. We conjecture this is because HS involves commonsense reasoning, where the vocabulary map is more crucial, while other datasets focus on mathematical reasoning, requiring stronger calculation abilities from LMs. Additionally, it is worth mentioning that **orthogonal techniques**, such as Self-Training with DPO (Wang et al., 2024a) and Self-Correction (SCORE) (Zhang et al., 2024b), significantly enhance SLMs' reasoning without LLMs. Since both of them involve fine-tuning SLMs, they could be **seamlessly incorporated** into our framework, where we leave i) the integration of DynSDPB with these methods and ii) the exploration of DynSDPB's potential to improve SLMs' reasoning abilities for future work.

Table 2: Results on comparison with KD. We report accuracy for all the datasets. † means from (Wang et al., 2023a) and †† means from (Ren et al., 2023). The results of GKD-CLS and ReAugKD are from (Wang et al., 2022b) and (Zhang et al., 2023), respectively.

| Method | Student | #Params | RTE | MRPC | MNLI-m/mm | SST-2 | QNLI | QQP |
|---|---|---|---|---|---|---|---|---|
| BERT-base$_{12}$ (Teacher)† | - | 109.0M | 69.3 | 85.5 | 84.1/83.1 | 92.7 | 90.5 | 89.2 |
| Finetune† (Devlin et al., 2018) | BERT-base$_6$ | 67.0M | 64.9 | 79.2 | 81.1/79.8 | 89.5 | 86.5 | 88.2 |
| Vanilla KD† (Hinton et al., 2015) | BERT-base$_6$ | 67.0M | 65.1 | 79.8 | 82.4/81.6 | 91.4 | 86.9 | 88.4 |
| PKD† (Sun et al., 2019) | BERT-base$_6$ | 67.0M | 65.5 | 79.9 | 81.5/81.0 | 92.0 | 89.0 | 88.9 |
| GKD-CLS (Wang et al., 2022b) | BERT-base$_6$ | 67.0M | - | - | 82.6/81.9 | 93.0 | 89.5 | 71.6 |
| Hard-Action KD† (Wang et al., 2023a) | BERT-base$_6$ | 67.0M | 66.0 | 82.2 | 82.6/81.8 | 92.1 | 89.0 | 88.9 |
| Soft-Action KD† (Wang et al., 2023a) | BERT-base$_6$ | 67.0M | 66.8 | 82.2 | 83.1/82.1 | 92.6 | 89.3 | 89.1 |
| Meta Distill†† (Zhou et al., 2021) | BERT-base$_6$ | 67.0M | 65.6 | 79.5 | 82.4/81.4 | 92.9 | 88.9 | 88.5 |
| LGTM†† (Ren et al., 2023) | BERT-base$_6$ | 67.0M | 67.4 | 83.3 | 83.4/82.5 | 93.4 | 90.2 | 89.3 |
| ReAugKD (Zhang et al., 2023) | BERT-base$_6$ | 67.0M | 70.4 | 86.3 | -/- | 92.5 | 90.7 | 91.2 |
| Sequential DLB (Without Teachers) | BERT-base$_6$ | 67.0M | 66.5 | 81.5 | 81.7/81.5 | 90.5 | 87.1 | 89.9 |
| Random DLB (Without Teachers) | BERT-base$_6$ | 67.0M | 67.5 | 82.1 | 82.2/81.9 | 90.9 | 87.8 | 90.2 |
| **DynSDPB (Ours)** (Without Teachers) | BERT-base$_6$ | 67.0M | 68.2 | 82.6 | 82.7/82.2 | 91.5 | 88.4 | 91.0 |

Table 3: Results on NLU tasks from the Super-GLUE (Wang et al., 2019) benchmark. The best and second-best results for each group of student models are in **bold** and *italics*, respectively.

| Methods | Counterpart | BoolQ | CB | COPA | RTE | WiC |
|---|---|---|---|---|---|---|
| Finetune | DeBERTa-large | 63.6 | 71.4 | 58.0 | 82.3 | 69.4 |
| Double Finetune | DeBERTa-large | 64.7 | 80.4 | 65.0 | 83.1 | 73.5 |
| Sequential DLB | DeBERTa-large | 64.5 | 81.4 | 66.0 | 85.9 | 73.7 |
| Random DLB | DeBERTa-large | *64.9* | *83.9* | *67.0* | *86.6* | *73.8* |
| **DynSDPB (Ours)** | DeBERTa-large | **65.6** | **85.7** | **68.0** | **88.1** | **74.3** |
| Finetune | DeBERTa-v2-large | 63.2 | 73.4 | 71.0 | 83.4 | 71.6 |
| Double Finetune | DeBERTa-v2-large | 64.9 | 74.6 | 65.0 | 85.1 | 73.9 |
| Sequential DLB | DeBERTa-v2-large | 65.5 | 82.3 | 80.0 | 86.9 | 74.3 |
| Random DLB | DeBERTa-v2-large | *67.3* | *84.1* | *82.0* | *87.8* | *74.9* |
| **DynSDPB (Ours)** | DeBERTa-v2-large | **68.7** | **86.2** | **84.0** | **90.2** | **75.4** |
| Finetune | DeBERTa-v3-large | 62.2 | 78.6 | 82.0 | 88.5 | 74.1 |
| Double Finetune | DeBERTa-v3-large | 67.4 | 84.0 | 85.0 | 90.1 | 74.9 |
| Sequential DLB | DeBERTa-v3-large | 68.8 | 83.9 | 83.0 | 89.5 | 75.2 |
| Random DLB | DeBERTa-v3-large | *69.1* | *85.7* | *86.0* | *90.6* | *75.9* |
| **DynSDPB (Ours)** | DeBERTa-v3-large | **70.1** | **87.5** | **87.0** | **91.7** | **76.7** |

Table 4: Results of fine-tuning LLaMA model families on NLG tasks. The best and second-best results for each group of student models are in **bold** and *italics*, respectively.

| Methods | Counterpart | GSM8K | SVAMP | HS | AQUA | MQA |
|---|---|---|---|---|---|---|
| Finetune | Llama-1-7B | 23.5 | 54.6 | 30.9 | 24.8 | 29.1 |
| Double Finetune | Llama-1-7B | 25.2 | 59.3 | 35.9 | 29.1 | 29.6 |
| Random DLB | Llama-1-7B | *26.1* | *60.1* | *50.1* | *29.7* | *30.1* |
| **DynSDPB (Ours)** | Llama-1-7B | **27.1** | **61.4** | **56.8** | **30.8** | **30.9** |
| Finetune | Llama-2-7B | 27.6 | 57.3 | 37.4 | 29.5 | 30.0 |
| Double Finetune | Llama-2-7B | 30.3 | 57.7 | 52.5 | 30.3 | 30.5 |
| Random DLB | Llama-2-7B | *31.3* | *58.3* | *70.1* | *31.1* | *30.9* |
| **DynSDPB (Ours)** | Llama-2-7B | **32.2** | **60.2** | **85.2** | **32.2** | **31.8** |
| Finetune | Llama-2-13B | 36.3 | 65.6 | 50.5 | 30.7 | 34.1 |
| Double Finetune | Llama-2-13B | 37.8 | 66.1 | 77.3 | 31.2 | 34.8 |
| Random DLB | Llama-2-13B | *43.4* | *67.2* | *81.4* | *34.2* | *35.5* |
| **DynSDPB (Ours)** | Llama-2-13B | **45.9** | **68.7** | **91.2** | **35.1** | **39.9** |
| Finetune | Llama-3-8B | 51.5 | 72.3 | 60.2 | 39.4 | 52.1 |
| Double Finetune | Llama-3-8B | 52.3 | 74.1 | 80.1 | 40.9 | 52.9 |
| Random DLB | Llama-3-8B | *57.1* | *74.6* | *85.1* | *42.1* | *53.1* |
| **DynSDPB (Ours)** | Llama-3-8B | **58.9** | **77.0** | **93.1** | **43.2** | **54.2** |

## 4.3 DISCUSSION

**Comparison with KD.** Since our method is strongly related to KD, here we provide a focused comparison to representative KD methods for PLMs introduced in Subsection 4.1's Baselines. Table 2 compares static/dynamic SelfD with some represenetive KD techniques where the teacher (12-layer-BERT-base (Devlin et al., 2018)) is pre-trained and provides posterior targets for the student (6-layer-BERT-base). As expected, some advanced KD approaches such as LGTM (Ren et al., 2023) and ReAugKD (Zhang et al., 2023) does remarkably improve students' performance for most datasets. However, the results from RTE, MRPC, and QQP show that self-teaching of students based on last iteration's logits can sometimes be more effective than being guided by a pre-trained teacher. Therefore, in practice, if there are no internet or sufficient budgets to deploy LLMs to help fine-tune students, applying DynSDPB is advisable to accomplish given downstream NLP tasks.

**Gradient Vanishing Mitigation.** To better understand DynSDPB's effectiveness, in Figure 2 we plot the two gradient norms of the loss function with respect to different layers of DeBERTa-large (He et al., 2020) on BoolQ (Clark et al., 2019), for vanilla fine-tuning and fine-tuning via dynamic SelfD, respectively. From Figure 2a, we see that large meaningful gradients only exist in the top layers and gradients start vanishing in the bottom layers at the beginning of training iteration 500.

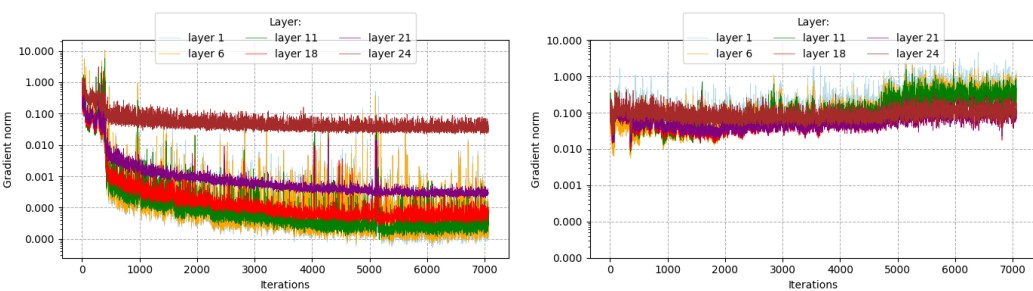

(a) Vanilla Fine-tuning BoolQ (Gradient Vanishing).

(b) Fine-tuning BoolQ via Dynamic SelfD.

Figure 2: The logarithmic-scale gradient norms of selected layers for DeBERTa-large fine-tuning in two ways. The gradients of all parameters within one layer are averaged into a scalar value, whose values' changes are tracked throughout fine-tuning iterations. We observe that for vanilla fine-tuning, the gradients of shallow layers vanish by the end of the process. However, the robust gradients always exist to benefit fine-tuning if applying dynamic SelfD.

This is in large contrast to Figure 2b, where we observe that robust gradients always exist during the whole fine-tuning process. Similar visualizations for DeBERTa-v3-large (He et al., 2021) and RoBERTa-base (Liu et al., 2019b) can be found in Figure 4 and Figure 5 in Appendix, respectively, where we observe similar behaviors of how gradients are changing as fine-tuning iterations go.

**Effectiveness of Information from the Last Mini-batch.** We conduct an ablation study to explore the effects of the proposed LMBC regularization loss. Both Table 1 and Table 3 show that LMs improve performance via SelfD from the last mini-batch compared with Finetune indicated by the "Sequential/Random DLB" rows. Moreover, we can see that Random DLB generally performs better than its Sequential counterpart. We conjecture that the underlying reason is due to shuffling, an effective mechanism to ensure that learning doesn't get biased or overfitted to specific patterns within the training data.

**How does the dynamic strategy help?** Table 1, Table 3, and Table 4 highlight that after adding our dynamic strategy, the final performance gets further boosted indicated by the comparison between "DynSDPB" rows and "Random DLB" rows, indicating that adaptive fine-tuning could somehow improve performance.

**What if we only apply dynamic strategy?** We run ablation experiments using only the dynamic strategy, called "Dynamic Finetune", where we rely solely on the uncertainty and discriminatory signals from the current mini-batch to dynamically adjust $L_{\mathrm{CE}}^{\theta_t}$ without the LMBC loss. This approach slightly improves performance compared to regular Finetune, but it is still worse than the "DynSDPB" method, indicated by the "Dynamic Finetune" row in Table 5.

Table 5: Ablation results on the dynamic strategy where "Dynamic Finetune" means using only the dynamic strategy without last mini-batch's information. The best results are in **bold**.

| Methods | Datasets | RoBERTa-base$_6$ | BERT-base$_6$ | ALBERT-base$_6$ | RoBERTa-base$_{12}$ | BERT-base$_{12}$ | ALBERT-base$_{12}$ |
|---|---|---|---|---|---|---|---|
| Finetune | RTE | 60.6 | 64.9 | 57.8 | 64.9 | 69.3 | 68.9 |
| Double Finetune | RTE | 61.7 | 63.9 | 61.1 | 73.3 | 69.7 | 70.7 |
| Dynamic Finetune | RTE | 61.2 | 64.5 | 61.5 | 72.5 | 69.9 | 71.3 |
| Random DLB | RTE | 67.6 | 67.5 | 65.1 | 79.1 | 71.1 | 74.4 |
| **DynSDPB (Ours)** | RTE | **68.3** | **68.2** | **67.2** | **79.8** | **71.9** | **75.8** |
| Finetune | COLA | 52.1 | 38.3 | 48.9 | 59.6 | 56.9 | 56.1 |
| Double Finetune | COLA | 53.7 | 38.6 | 49.4 | 61.5 | 57.6 | 56.5 |
| Dynamic Finetune | COLA | 54.1 | 39.5 | 49.9 | 61.1 | 57.8 | 57.0 |
| Random DLB | COLA | 55.0 | 42.8 | 51.1 | 62.2 | 58.9 | 58.2 |
| **DynSDPB (Ours)** | COLA | **56.0** | **42.5** | **51.9** | **62.8** | **59.7** | **69.4** |

**Hyperparameter Sensitivity Analysis.** Here we evaluate the heatmap in terms of temperature $\tau$ and $\alpha$ for sensitivity analysis. The results of 30 "Random DLB" experiments for DeBERTa-v3-large

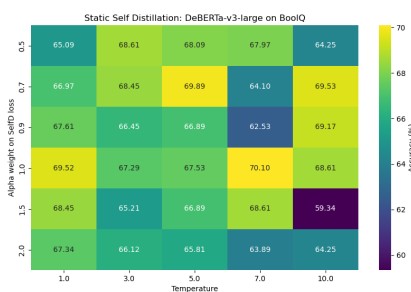 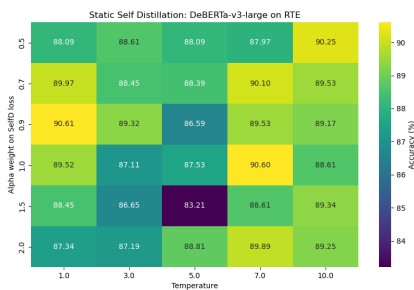

(a) The heatmap for DeBERTa-v3-large on BoolQ.    (b) The heatmap for DeBERTa-v3-large on RTE.

Figure 3: The heatmap evaluation on hyperparameters (temperature $\tau$ and balancing factor $\alpha$) for static SelfD (Random DLB) for DeBERTa-v3-large on BoolQ and RTE.

on BoolQ and RTE are visualized in Figure 3a and Figure 3b, respectively. Intuitively, $\alpha$ being close to 2.0 means that we put more "trust" on students' dark knowledge. From the results, it seem to indicate that we should not put too much faith in it (e.g., setting $\alpha$ to be 1.0 is enough). Moreover, we should carefully consider the mutual influence between $\tau$ and $\alpha$ in practice.

## 5    CONCLUSION

In this paper, we propose a SelfD method called **dyn**amic **S**elfD from the **p**revious mini-**b**atch (DynSDPB), which enables instant distillation via logits from the last-mini batch. To handle incorrect predictions in early iterations, we dynamically adjust the distillation influence and temperature for better fine-tuning. We also introduce Vocabulary Map Matching (VMM) to address the output dimension mismatch issues in auto-regressive LLMs. Moreover, DynSDPB enables seamless integration with existing self-correction and self-training methods for small language models (SLMs). We validate DynSDPB on encoder-only (e.g., BERT) and decoder-only (e.g., LLaMA) models, showing its effectiveness on both NLU and NLG tasks.

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

# A RELATED WORKS

## A.1 KNOWLEDGE DISTILLATION FOR PLMs

Knowledge distillation (KD) (Hinton et al., 2015) is widely used in computer vision (CV) to compress and accelerate deep neural networks (DNNs) such as ResNet-50 (He et al., 2016). Recently, researchers have attached great significance to the KD study in PLMs (Sun et al., 2022). Specifically, KD works about PLMs can be roughly classified into one-stage methods (task-specific) and two-stage ones (task-agnostic). **One-stage methods** perform distillation only at the fine-tuning stage, which is **task-specific**. For instance, Tang et al. (Tang et al., 2019) propose a KD method that distills BERT into a single-layer BiLSTM for some NLP tasks. BERT-PKD (Sun et al., 2019) performs KD with the teacher's logits and hidden states, and PD (Turc et al., 2019) is a novel KD pipeline that people could first pre-train compact models (six-layer BERT) with unlabeled text data and then explore transferring task-specific knowledge from large fine-tuned models (12-layer BERT) via standard KD. Further, numerous techniques have been proposed to enhance one-stage methods, which include applying multi-task learning (Liu et al., 2019a) or the mixup augmentation strategy (Liang et al., 2020), utilizing teachers' gradients (Wang et al., 2022b), employing multiple teacher models (Yuan et al., 2021; Wu et al., 2021a), dynamically adjusting three aspects (teacher model adoption, data selection, and KD objective adaptation) (Li et al., 2021), and adaptively selecting knowledge from teachers to transfer (Wang et al., 2023a). MetaDistil (Zhou et al., 2021) is the first to utilize meta learning to let teacher networks learn to better transfer knowledge to student networks via students' feedback. Liu et al. (Liu et al., 2022) propose Multi-Granularity Structural KD that utilizes intermediate representations from multiple semantic granularities (e.g., tokens, spans and samples). Wu et al. (Wu et al., 2023) explore the token-level attribution-based knowledge to improve knowledge transfer. Yang et al. (Yang et al., 2022) propose a sparse teacher trick to remove the parameters resulting in student unfriendlines under the guidance of an overall knowledgeable score. Recently, inspired by MetaDistil (Zhou et al., 2021), Ren et al.(Ren et al., 2023) propose LGTM that can efficiently incorporate distillation influence into the teacher's learning process to better guide the student. Moreover, Sengupta et al. (Sengupta et al., 2023) point out the drawbacks of MetaDistil (Zhou et al., 2021) and introduce a meta-policy KD framework called MPDistil. **Two-stage methods** perform distillation at both the pre-training stage and the fine-tuning stage, which is usually **task-agnostic**. For instance, DistilBERT (Sanh et al., 2019), MINILM (Wang et al., 2020b), and MobileBERT (Sun et al., 2020) all focus on the pre-training stage, aiming to get a lightweight task-agnostic model that can be fine-tuned on unknown downstream tasks. MINILMv2 (Wang et al., 2020a) uses self-attention relation distillation to generalize and simplify MINILM (Wang et al., 2020b). TinyBERT (Jiao et al., 2019) performs Transformer distillation at both the pretraining and task-specific learning stage to further improve KD performance. Wu et al. (Wu et al., 2021b) show that it is beneficial to augment KD with a third objective that encourages the student to imitate the causal dynamics of the teacher through a distillation interchange intervention training objective (DIITO). HomoDistil (Liang et al., 2023) equipped with iterative pruning could beat existing task-agnostic baselines. Recently, Dasgupta et al. (Dasgupta et al., 2023) propose a novel KD loss that is agnostic to both architecture and tasks based on Taylor Series Expansion of the Loss. Interestingly, Lee et al. (Lee et al., 2023) even study Distillation from Weak Teacher (DWT) for the PLMs' pre-training stage. **Last but not least**, there are some works that can be both applied to enhance task-specific distillation and finetuning task-agnostic distilled models. For example, Park et al. (Park et al., 2021) propose a KD objective that transfers the contextual knowledge via two types of relationship (Word Relation and Layer Transforming Relation). ReAugKD (Zhang et al., 2023) implements a flexible KD method via a Retrieval-augmented KD framework. MINIDISC (Zhang et al., 2024a) is an efficient method to scheduling an optimal teacher assistant to bridge the capacity gap between the teacher and the student. **All the previous works** and the **scope of this paper** focuses on the **encoder-only** language models. MiniLLM (Gu et al., 2023) is the first work that uses a white-box KD method to distill LLMs for text generation tasks. Based on MiniLLM, works on studying KD for auto-regressive LLMs (Agarwal et al., 2024; Ko et al., 2024) have recently attracted researchers' attention.

In summary, all the methods above could be categorized under the **standard KD methodology** because they all involve pre-training the large teacher model first and then fine-tuning the student model, which is frequently time-consuming and computation-expensive. In this paper, we explore a scenario in which the student model distills knowledge within itself instead of approximating the

output logits from a pre-trained teacher model because sometimes the experienced teacher model might not be available or feasible due to computational or storage constraints.

## A.2 SELF-DISTILLATION IN PLMS

It's worth noting that a method known as Self-distillation (SelfD) (Furlanello et al., 2018; Yang et al., 2019; Zhang et al., 2019; Yun et al., 2020; Zheng & Peng, 2022; Shen et al., 2022) exists, which is a particular form of KD where the teacher and student have the same architecture, typically belonging to a special type of KD. However, all of the previous SelfD papers focus on the CV domain. Hahn et al. (Hahn & Choi, 2019) propose a method called self-knowledge distillation based on the soft target probabilities of the training model itself, which is the first paper focusing on two NLP tasks: language model and neural machine translation. With the emergence of PLMs (Sun et al., 2022), people have begun to study the application of SelfD on them. Interestingly, Early Exit (EE) (Xu & McAuley, 2023) in BERT (Devlin et al., 2018) resembles one SelfD work in CV (Zhang et al., 2019), aiming for accelerating PLM inference by stopping it at a specific Transformer layer based on predefined criteria. Though not reducing model sizes, it decreases computation by using inserted internal classifiers into a Transformer-based model (e.g., 12-layer BERT-base). This is similar to (Zhang et al., 2019), which adds extra classifiers into ResNets' (He et al., 2016) intermediate layers for SelfD training, thus enhancing models' generalization capability. EE techniques for PLMs focus on **exit criteria**, which currently have **three types** (Xu & McAuley, 2023): confidence estimation, internal ensemble, and learning to exit.

The **first** technique is **Confidence Estimation**. Certain works in CV (Park et al., 2015; Teerapittayanon et al., 2016) define a metric as a confidence proxy for predictions, where inference can terminate early if this confidence metric surpasses a preset threshold in early layers. DeeBERT (Xin et al., 2020) is the first work that applies this concept to PLMs, where linear internal classifiers (ICs) are added after each Transformer layer. During inference, the model exits early when an IC predicts a probability with an entropy below the threshold. A similar strategy is adopted in RightTool (Schwartz et al., 2020), using temperature-calibrated maximum class probability as confidence. FastBERT (Liu et al., 2020) employs the idea of SelfD, distilling the final classifier's output into earlier classifiers for improved performance. Subsequently, RomeBERT (Geng et al., 2021) introduces gradient regularization to aid SelfD with the purpose of ameliorating DeeBERT (Xin et al., 2020). SkipBERT (Wang et al., 2022a) replaces lower BERT layers with pre-computed text chunk representations and implements confidence-based EE for higher layers, achieving maximal acceleration.

The **second** one is **Internal Ensemble**. Confidence estimation suffers from poor utilization of computation when an IC's confidence doesn't meet the exit criteria, rendering finished computational work meaningless and invalid. Utilizing results from preceding layers to enhance EE quality is a promising research direction. Internal Ensemble methods leverage outputs and predictions from multiple internal classifiers for better decision-making. The pioneering work, PABEE (Zhou et al., 2020), draws a comparison between overfitting in training and overthinking in inference and lets the model exit when consecutive ICs produce unchanged predictions. Sun et al. (Sun et al., 2021) introduce a novel objective function for the training of the ensemble ICs and utilize a voting mechanism for internal ensemble decisions. LeeBERT (Zhu, 2021) enhances IC prediction through mutual distillation and follows PABEE's patience-based exiting strategy (Zhou et al., 2020). Liao et al. (Liao et al., 2021) introduce a global past-future perspective for the ensemble ICs' predictions. PCEE-BERT (Zhang et al., 2022) combines patience-based exiting with confidence estimation and terminates inference when enough consecutive intermediate layers are confident about their predictions.

The **third** one, **Learning to Exit**, employs a learning-based strategy for exit decisions. BERxiT (Xin et al., 2021) trains a linear Learning-to-Exit (LTE) module to forecast the accuracy of the current internal IC's predictions. CAT (Schuster et al., 2021) trains additional prediction heads on top of intermediate layers and dynamically decides when to stop allocating computational effort to each input via a meta consistency classifier. MPEE (Kong et al., 2022) is a multi-perspective framework where the vertical architecture uses recycling EE classifier memory and weighted SelfD to enhance ICs and the horizontal perspective uses recycling class attention memory to emphasize the informative tokens.

To sum up, all the existing EE methods mentioned above can be integrated into a **unified framework**: Given a 12-layer-BERT-base model (acting as the teacher in KD), additional classifiers are consecutively attached to each Transformer layer, and the entire model is fine-tuned together. At inference time, a sample can perform EE via one of the intermediate classifiers based on various exit criteria. The **core goal** of EE is to speed up the inference process of the original model (e.g., 12-layer BERT-base model). However, this framework has **two main limitations**. **Firstly**, similar to KD, it presupposes the existence of a proficient teacher model. Unlike KD that allows a student model to learn from the teacher, EE generally modifies the teacher model's architecture to increase inference speed and reduce computational costs. **Moreover**, those methods assume that the given PLM is fully open-source, offering people access to training methods, dataset specifics, model weights, and **crucially**, modifications to the architecture of the original model (e.g., adding extra classifiers to the intermediate layers). However, in practice, numerous PLMs and LLMs are closed-source, which limits the applicability of existing EE techniques. In this work, we propose a SelfD method that involves merely altering the data loading manner for different downstream tasks so that we don't have to care whether the given LM is open-source or not.

## B ALGORITHM

---

**Algorithm 1** Pseudo code for DynSDPB.

---

1: **Input:** balancing coefficient $\alpha$
2: **Input:** distillation temperature $\tau$
3: **Require:** a special data_loader
4: last_logits = None ▷ Initialization of last mini-batch's information
5: **for** (x, labels) in data_loader **do**
6:     $n$ = x.size(0) ▷ Batch size of the current iteration
7:     logits = model.forward(x)
8:     loss = CE_Loss(logits, labels)
9:     **if** last_logits $\neq$ None **then**
10:         $\tilde{\tau} = d_x \cdot \tau$ ▷ Re-scale $\tau$ via discrimination capability $d_{x_i}$
11:         $\tilde{\alpha} = (1 - \frac{u_{x_i}}{U}) \cdot \alpha$ ▷ Re-scale $\alpha$ via prediction uncertainty $u_{x_i}$
12:         soft_targets = Softmax(last_logits/$\tilde{\tau}$)
13:         pred = Softmax(logits[:$\hat{n}$/2]/$\tilde{\tau}$)
14:         loss += $\tilde{\alpha}\cdot$ KL_Loss(soft_targets, pred)
15:     **end if**
16:     loss.backward() ▷ update parameters
17:     last_logits = logits[:$\hat{n}$/2].detach()
18: **end for**

---

## C DATASETS

### C.1 NLG DATASETS

#### C.1.1 GLUE BENCHMARK

The General Language Understanding Evaluation (GLUE) benchmark (Wang et al., 2018) is a collection of diverse natural language understanding tasks, including Multi-Genre Natural Language Inference (MNLI) (Williams et al., 2017), Quora Question Pairs (QQP) (Chen et al., 2018), Question Natural Language Inference (QNLI) (Rajpurkar et al., 2016), Stanford Sentiment Treebank (SST-2) (Socher et al., 2013), Microsoft Research Paraphrase Corpus (MRPC) (Dolan & Brockett, 2005), Recognizing Textual Entailment (RTE) (Bentivogli et al., 2009), and Corpus of Linguistic Acceptability (COLA) (Warstadt et al., 2019), which we use in this paper. It serves as a benchmark for evaluating the performance of models across various language understanding tasks.

**Multi-Genre Natural Language Inference (MNLI)** MNLI (Williams et al., 2017) is a task where models are required to determine the logical relationship between a pair of sentences, classifying them into one of three categories: entailment, contradiction, or neutral. Its test and development

datasets are further divided into in-domain (MNLI-m) and cross-domain (MNLI-mm) splits to evaluate the generality of tested models. The number of training size is approximately 392,702 pairs, validation size is 20,000 pairs, and test size is 20,000 pairs.

**Quora Question Pairs (QQP)**  QQP (Chen et al., 2018) involves determining whether pairs of questions posted on Quora are semantically equivalent or not. This task is framed as a binary classification problem. The number of training size is approximately 363,860 pairs, validation size is 40,000 pairs, and test size is 390,965 pairs.

**Question Natural Language Inference (QNLI)**  QNLI (Rajpurkar et al., 2016) is similar to MNLI but focuses specifically on question answering. Models must determine if a given sentence answers a given question, categorizing the relationship between them as entailment, contradiction, or neutral. The number of training size is approximately 104,743 pairs, validation size is 5,700 pairs, and test size is 5,800 pairs.

**Stanford Sentiment Treebank (SST-2)**  SST-2 (Socher et al., 2013) is a sentiment analysis task where models are tasked with classifying the sentiment of a given sentence as positive or negative. The number of training size is approximately 67,349 pairs, validation size is 872 pairs, and test size is 1,821 pairs.

**Microsoft Research Paraphrase Corpus (MRPC)**  MRPC (Dolan & Brockett, 2005) involves identifying whether pairs of sentences are paraphrases of each other or not. Like QQP, this task is framed as a binary classification problem. The number of training size is approximately 3,668 pairs, validation size is 408 pairs, and test size is 1,725 pairs.

**Recognizing Textual Entailment (RTE)**  RTE (Bentivogli et al., 2009) requires determining whether a given hypothesis can be inferred from a given piece of text. Models must classify the relationship between the text and the hypothesis as either entailment or not entailment. The number of training size is approximately 2,490 pairs, validation size is 277 pairs, and test size is 3,000 pairs.

**Corpus of Linguistic Acceptability (COLA)**  COLA (Warstadt et al., 2019) is a linguistic acceptability judgment task where models are tasked with determining whether a given sentence is grammatically and semantically acceptable in English. This task is typically framed as binary classification. The number of training size is approximately 8,550 pairs, validation size is 1,045 pairs, and test size is 1,045 pairs.

### C.1.2 SUPERGLUE BENCHMARK

SuperGLUE (Super General Language Understanding Evaluation) (Wang et al., 2019) is a benchmark suite designed to evaluate and improve the performance of ML models on a variety of natural language understanding tasks. It was introduced as a more challenging successor to the original GLUE benchmark (Wang et al., 2018), reflecting the rapid advancements in NLP technologies and model capabilities. In this paper, we use Boolean Questions (BoolQ) (Clark et al., 2019), CommitmentBank (CB) (De Marneffe et al., 2019), Choice of Plausible Alternatives (COPA) (Roemmele et al., 2011), and Word-in-Context (WiC) (Pilehvar & Camacho-Collados, 2018) tasks to evaluate our method.

**Boolean Questions (BoolQ)**  BoolQ (Clark et al., 2019) is a reading comprehension task requiring models to answer yes/no questions based on short passages. It comprises of binary questions using the Google search engine as their source of questions; they are then paired with appropriate paragraphs from Wikipedia articles that contain the relevant answers. The number of training size is approximately 9,247 pairs and validation size is 3,270 pairs.

**CommitmentBank (CB)**  CB (De Marneffe et al., 2019) comprises of short texts with embedded clauses. The examples are taken from sources like British National Corpus Fiction and Wall Street Journal. It involves a three-class textual entailment task. Each example includes a premise and the corresponding hypothesis along with the class label "contradiction", "neutral", or "entailment". The

number of training size is approximately 250 pairs, validation size is 56 pairs, and test size is 250 pairs.

**Choice of Plausible Alternatives (COPA)**  COPA (Roemmele et al., 2011) is a causal reasoning task which involves selecting the most plausible choice for a cause or effect given a premise. The number of training size is approximately 400 pairs, validation size is 100 pairs, and test size is 500 pairs.

**Word-in-Context (WiC)**  WiC (Pilehvar & Camacho-Collados, 2018) is a task focused on word sense disambiguation, comprising of binary classification of pairs of sentences. In this task, two text snippets are provided, each containing a word that could have multiple meanings. The goal is to ascertain whether the word has the same meaning in both sentences. The number of training size is approximately 6,000 pairs, validation size is 638 pairs, and test size is 1,400 pairs.

## C.2   NLG DATASETS

For mathematical tasks, we select four datasets. GSM8K (Cobbe et al., 2021) is a high-quality linguistically diverse dataset of grade school math word problems. SVAMP (Patel et al., 2021) contains simple math word problems created by applying carefully chosen variations to examples sampled from existing datasets. The AQUA dataset is designed to test and train models on mathematical problem-solving, particularly focused on algebra and arithmetic. MathQA (Amini et al., 2019) is an advanced dataset gathered by using a new representation language to annotate the AQUA dataset (Ling et al., 2017) with fully specified operational programs. For commonsense tasks, we select HellaSwag (HS) (Zellers et al., 2019). HS is a challenging dataset, which contains questions to select the best endings to complete sentences. It has been considered as one of the most common datasets to judge the reasoning ability of LLMs.

**Grade School Math 8K (GSM8K)**  GSM8K (Cobbe et al., 2021) is a dataset specifically designed to challenge language models with grade-school level math problems. These problems require both arithmetic and logical reasoning to solve. The dataset is intended for evaluating the mathematical reasoning capabilities of language models. It contains approximately 8,500 problem-answer pairs.

**Synthetic Variable Arithmetic Math Problems (SVAMP)**  SVAMP (Patel et al., 2021) is a dataset composed of math word problems that require understanding of arithmetic operations and the ability to deal with variable quantities. This dataset is designed to test both comprehension and arithmetic skills in a more controlled synthetic setting. It contains about 1,000 annotated problem-solution pairs.

**AQUA**  The AQUA (Algebra Question Answering) Ling et al. (2017) dataset in the context of NLP is designed to test and train models on mathematical problem-solving, particularly focused on algebra and arithmetic. It contains word problems that typically require algebraic reasoning, and each problem is accompanied by multiple-choice answers (typically A, B, C, D, etc.). It contains around 100,000 algebraic word problems, which cover a range of difficulty levels.

**MathQA**  MathQA (Amini et al., 2019) is a large-scale dataset of math word problems and their corresponding solutions. It covers a range of topics from algebra to geometry. The dataset not only provides the problems and solutions but also includes multiple choice answers and detailed reasoning steps. It contains over 37,000 problem-answer pairs.

**HellaSwag (HS)**  HS (Zellers et al., 2019) is a dataset aimed at testing commonsense reasoning and abductive reasoning within natural language understanding models. It presents contexts from a wide array of domains and requires models to predict the most likely or plausible continuation among given choices. It includes around 70,000 context-completion pairs.

## D  IMPLEMENTATION DETAILS

We run all experiments on GeForce RTX 4090 or A6000. For hyperparameter selections, we do gird search of learning rate $lr$, $\alpha$ and $\tau$ similar to the strategy proposed by Sun et al. (Sun et al., 2019).

**NLU tasks.** For Sequential/Random DLB, we perform grid search for $lr \in \{5 \times 10^{-6}, 1 \times 10^{-5}, 2 \times 10^{-5}, 3 \times 10^{-5}\}$, $\alpha \in \{0.5, 0.7, 0.9, 1.0, 1.5, 2.0\}$ and $\tau \in \{1, 3, 5, 7, 10\}$. For DynSDPB, we perform grid search for $lr \in \{5 \times 10^{-6}, 1 \times 10^{-5}, 2 \times 10^{-5}, 3 \times 10^{-5}\}$, $\alpha \in \{0.2, 0.3, 0.4, 0.6, 0.8, 1.0\}$ and $\tau \in \{1, 3, 5\}$. Then, we choose the combination that performs best on the validation set. Additionally, we initialize student models with the embedding layer and first 6 hidden layers of the original model such as 12-layer BERT-base if we want to have some 6-layer variants. We set the fine-tuning epoch number as 6 for Double Finetune and 3 for others methods. For the GLUE (Wang et al., 2018) tasks, we fix the training batch size as 32 and validation batch size as 64. For the SuperGLUE (Wang et al., 2019) tasks, we fix the training batch size as 8 and validation batch size as 8. Moreover, we use PyTorch's AdamW and linear schedule (Imambi et al., 2021) to do stochastic gradient descent (SGD).

**NLG tasks.** For both Random DLB and DynSDPB, we perform grid search for $lr \in \{1 \times 10^{-4}, 1.5 \times 10^{-4}, 2 \times 10^{-4}\}$, $\alpha \in \{0.5, 0.7, 0.9\}$ and $\tau \in \{3, 5, 7\}$. Then, we choose the combination that performs best on the validation set. We use LoRA (Hu et al., 2021) to fine-tune all decoder-only SLMs. For all experiments, we follow the setting (Shi et al., 2024) where we set the rank $r = 4$, $\alpha = 8$, the fine-tuning epoch number as 40, the training batch size as 2, and the validation batch size as 8.

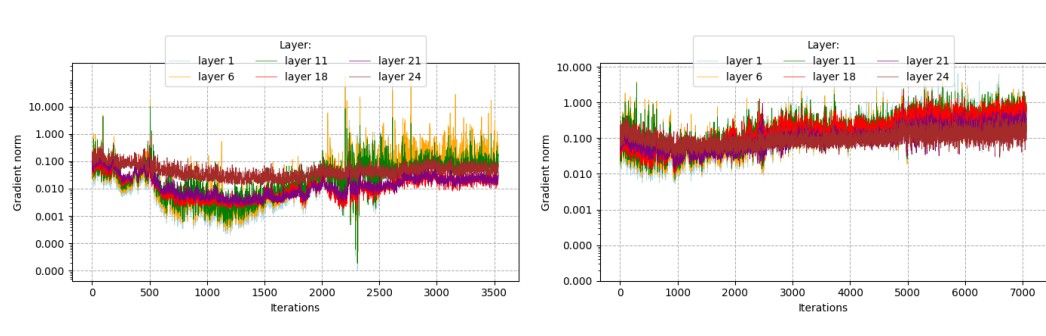

(a) Vanilla Fine-tuning BoolQ (Gradient Vanishing)   (b) Fine-tuning BoolQ via Dynamic SelfD

Figure 4: The logarithmic-scale gradient norms of selected layers for DeBERTa-v3-large fine-tuning in two ways. The gradients of all parameters within one layer are averaged into a scalar value, whose values' changes are tracked throughout fine-tuning iterations. We observe that for vanilla fine-tuning, the gradients of shallow layers vanish by the end of the process. However, the robust gradients always exist to benefit fine-tuning if applying dynamic SelfD.

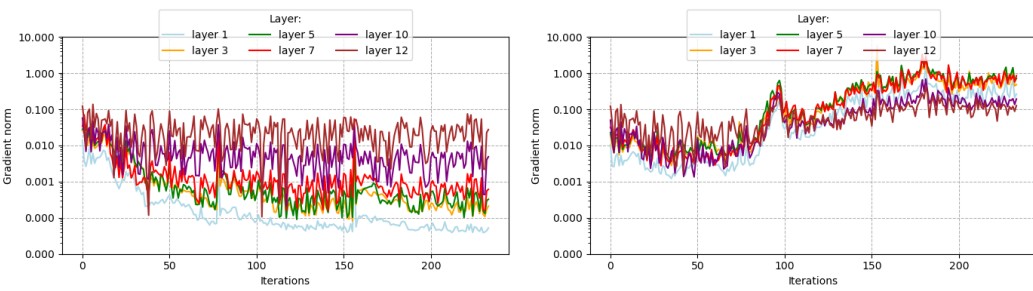

(a) Vanilla Fine-tuning RTE (Gradient Vanishing)   (b) Fine-tuning RTE via Dynamic SelfD

Figure 5: The logarithmic-scale gradient norms of selected layers for RoBERTa-base fine-tuning in two ways. The gradients of all parameters within one layer are averaged into a scalar value, whose values' changes are tracked throughout fine-tuning iterations. We observe that for vanilla fine-tuning, the gradients of shallow layers vanish by the end of the process. However, the robust gradients always exist to benefit fine-tuning if applying dynamic SelfD.

