# OpenReview forum: "Dynamic Self-Distillation via Previous Mini-batches for Fine-tuning Small Language Models"
_ICLR.cc/2025/Conference — ICLR 2025 Conference Withdrawn Submission_

### Official Review · Reviewer_RAEi · 2024-10-29

**Soundness:** 3
**Presentation:** 1
**Contribution:** 2
**Rating:** 3
**Confidence:** 4

**Summary:**

The work explores techniques of self-distillation in which logits from the latest mini-batch are exploited to supervise (regularize) the logits of the current batch.  Empirical studies are conducted with both encoder-only and decoder-only LMs on NLU and NLG tasks.

Here are my concerns:
1. Limited technical contributions.  The proposed method bears significant similarity with the existing work [Shen et al., Self-Distillation from the Last Mini-Batch for Consistency Regularization. In CVPR'22]. The only difference lies in dynamic adaption of the hyper-parameters. While the authors have emphasized the importance of dynamic hyper-parameter and empirical results seem verify the claim, I still think the innovations are limited given the existing work. Moreover, given the existing CVPR paper, it is not appropriate to claim that "To the best of our knowledge, we are the first to propose a SelfD method called DynSDPB....". Perhaps, you are the first to apply the technique in the field of NLP.
2. Poor organization. Related Work is an important part of an academic paper. Normally, we cannot leave related work to appendix. I also suggest the authors conduct proof-reading, as I have found many typos like the repetition of inspiration in Line 62.
3. More analysis are needed. It is better to show how the model gradually get improved in the process of self-distillation. It is unconvincing to just show the final performance and claim that the proposed method is effective.

**Strengths:**

1. Good performance on a wide range of tasks.

**Weaknesses:**

1. Limited technical innovations.
2. Poor presentation.

**Questions:**

N/A

---

### Official Review · Reviewer_c2CW · 2024-10-30

**Soundness:** 2
**Presentation:** 2
**Contribution:** 2
**Rating:** 3
**Confidence:** 4

**Summary:**

Knowledge distillation is a widely used method to reduce the computational cost of large language models (LLMs). However, the standard KD usually require an additional complex teacher model. Therefore, this paper focus on how to improve self-distillation in LLMs. Specifically, considering that the original self-distillation method is static, this paper introduces a dynamic SelfD method (DynSDPB) to address prediction inaccuracies during the early iterations. It dynamically adjusts the distillation influence and temperature value to enhance the adaptability of fine-tuning. Moreover, a vocabulary map matching is proposed to address the output inconsistency of the proposed method. Experimental results demonstrate the effectiveness of the proposed method.

**Strengths:**

**Strengths**
1. This paper introduces an adaptive method to dynamically adjust the influence based on the current states.
2. The proposed method can be a plug-in technique to integrated into existing self-training method.
3. Experimental results demonstrate the effectiveness of the proposed method.

**Weaknesses:**

**Weaknesses**
1. The main idea of this paper is just an extension of the original SelfD, which makes it contributions as incremental.
2. It seems authors only consider classification cases in NLU tasks. As authors claim their method as a task-agnostic method, so can you discuss how to adapt this method into regression tasks in NLU?
3. Authors claim that at the early stage, model prediction could be incorrect and affect self-distillation training. So why not conduct self-distillation when the training / model prediction is stable? Can you compare your method with a baseline that applies self-distillation after some initial training and discuss the trade-off between these approaches?
4. Why we need Output Mismatch Alignment? For decoder-only tasks, if the inputs are same, we only need to consider its logits and why we need to consider output mismatch problem during the fine-tuning? Could you provide some examples or explanations of when output mismatch occurs in practice during fine-tuning of decoder-only models.
5. The results from some baselines are mismatched from the original paper (e.g., RoBERTa and DeBERTa). Please see my comments in questions. Could you explain these discrepancies or the experimental setups that cause these differences.

**Minor issues**
1. Line 62: ''taking inspiration inspiration from DLB'' $\rightarrow$ ''taking inspiration from DLB''.

**Questions:**

1. How to apply the proposed method to regression tasks in NLU tasks (e.g., STS-B).
2. Some baselines seem too weak. For example, the original baselines in RoBERTa [a] and DeBERTa [b]:

|   | MNLI | QNLI  | QQP | RTE | SST | MRPC | CoLA |
| - | ------- | ----      | ------  | ----- | ----- | -------| ------|
| RoBERTa$_{base}$ | 87.6 | 92.8    | 91.9 | 78.7 | 94.8 | 90.2 | 63.6 |
| DeBERTa$_{large}$ | 91.1 |95.3 | 92.3 | 88.3 | 96.8 | 91.9 | 70.5 |
| DeBERTaV3$_{large}$ | 91.8| 96.0 | 93.0 | 92.7 | 96.9 | 92.2 | 95.3 |

It seems the baseline in this paper like (RTE, CoLA, SST, MRPC) are weaker than the original baseline. Please check these results.

[a] RoBERTa: A Robustly Optimized BERT Pretraining Approach.

[b] DeBERTaV3: Improving DeBERTa using ELECTRA-Style Pre-Training with Gradient-Disentangled Embedding Sharing.

---

### Official Review · Reviewer_dFip · 2024-11-04

**Soundness:** 2
**Presentation:** 2
**Contribution:** 2
**Rating:** 3
**Confidence:** 5

**Summary:**

This paper propose a model-agnostic method soft-label based self-distillation method in NLP. Since in NLP, there seems not much recent work about self-distillation, this work reference DLB (in CV) which utilize previous mini-batch for self-distillation and further adjust two hyper-parameter dynamically (temparature and CE/KL scale). Experiments on many NLU and NLG tasks shows improved performance over standard finetune and knowledge distillation-related methods.

**Strengths:**

1. Self-distillation in NLP remains under-explored. Recent soft-label-based knowledge distillation do require a large teacher model, in which either online KD or storing the probability need additional cost. This paper show Self-distillation would be a promising way to imporve the performance while remain low cost.

**Weaknesses:**

1. Novelty is limited as an incremental method paper. Self-distillation using information from previous batch is an idea from DLB[1]. This paper only add a dynamic strategy about changing temparature and scale during distillation.
2. This paper argue that DLB using fixed τ and α, which is not good. However, there lacks analysis on how dynamically change such hyper-parameters brings benefit (i.e. how this two parameters changed, and why this improve the performance intuitively)
3. For decoder-only models, there is a lack comparison to recent soft-label-based distillation methods.
4. For fine-tuning, it would be better to do full finetuning instead of Lora tuning. Lora tuning do not have comparable results to fine-tuning, and lora requires fewer compuation cost than your method I think, so your method would have an advantage over lora tuning which is expected. If the training resources does not support on large model, I woudl suggest to do full fine-tuning on small models such as 1B/3B.


[1]Self-distillation from the last mini-batch for consistency regularization. Yiqing Shen, Liwu Xu, Yuzhe Yang, Yaqian Li, and Yandong Guo.In Proceedings of the IEEE/CVF Conference on Computer Vision and Pattern Recognition, pp. 11943–11952, 2022.

**Questions:**

1. line 62: duplicated word "inspiration"
2. It would be better to add an overall benchmark score for each method so that the overall improvement can be easily checked.

---

### Note · Authors · 2024-11-19

I have read and agree with the venue's withdrawal policy on behalf of myself and my co-authors.